# Reassessing Fairness:
# A Reproducibility Study of NIFA's Impact on GNN Models

**Ruben Figge**[*]                                                    *ruben.figge@students.uva.nl*
*University of Amsterdam*

**Sjoerd Gunneweg**[*]                                            *sjoerd.gunneweg@student.uva.nl*
*University of Amsterdam*

**Aaron Kuin**[*]                                                    *aaron.kuin@student.uva.nl*
*University of Amsterdam*

**Mees Lindeman**[*]                                            *mees.lindeman@student.uva.nl*
*University of Amsterdam*

**Reviewed on OpenReview:** *https://openreview.net/forum?id=l5fXUKi8GO*

## Abstract

Graph Neural Networks (GNNs) have shown strong performance on graph-structured data but raise fairness concerns by amplifying existing biases. The Node Injection-based Fairness Attack (NIFA) (Luo et al., 2024) is a recently proposed gray-box attack that degrades group fairness while preserving predictive utility. In this study, we reproduce and evaluate NIFA across multiple datasets and GNN architectures. Our findings confirm that NIFA consistently degrades fairness—measured via Statistical Parity and Equal Opportunity—while maintaining utility on classical GNNs. However, claims of NIFA's superiority over existing fairness and utility attacks are only partially supported due to limitations in baseline reproducibility. We further extend NIFA to accommodate multi-class sensitive attributes and evaluate its behavior under varying levels of graph homophily. While NIFA remains effective in multi-class contexts, its impact is more sensitive in mixed and highly homophilic graphs. Although this is not a comprehensive validation of all NIFA claims, our work provides targeted insights into its reproducibility and generalizability across fairness-sensitive scenarios. The codebase is publicly available at: https://github.com/sjoerdgunneweg/Reassessing-NIFA.

## 1 Introduction

Graph Neural Networks (GNNs) (Gori et al., 2005) have achieved strong performance in tasks involving graph-structured data, with applications in domains such as computer vision, recommendation systems, and social network analysis (Wu et al., 2020). However, fairness remains a critical challenge, as GNNs can amplify biases present in the training data. This poses risks in high-stakes applications such as recruitment, credit scoring, law enforcement, and healthcare, where biased predictions can lead to unequal and potentially harmful outcomes (Dai & Wang, 2021).

Several recent works have explored fairness attacks on GNNs (Hussain et al., 2022; Kang et al., 2023; Zhang et al., 2024b), most of which rely on modifying the graph structure by altering edges between existing nodes. While effective, such interventions may not be realistic in practice due to security or access constraints. To address this, Luo et al. (2024) introduced the Node Injection-based Fairness Attack (NIFA), a gray-box method that injects synthetic nodes into the graph during training. NIFA aims to degrade group fairness

---

[*]Equal contribution

metrics—specifically Statistical Parity and Equal Opportunity—while maintaining utility metrics such as accuracy.

In this study, we critically evaluate the main claims of Luo et al. (2024) by reproducing their experiments across multiple datasets and GNN architectures. We also extend their work in two directions: (1) by generalizing NIFA to support multi-class sensitive attributes, and (2) by assessing its effectiveness under varying levels of graph homophily. Our reproduction confirms NIFA's fairness-degrading effect on classical GNNs with minimal utility loss, consistent with the original findings. However, results for fairness-aware GNNs diverge in two of the three evaluated cases, and the claim that NIFA outperforms existing attacks is only partially supported due to limited reproducibility of baselines.

Our extensions demonstrate that NIFA remains effective in multi-class fairness settings, but its impact possibly varies with graph structure. In particular, graphs with high homophily or mixed homophilic-heterophilic patterns—where connections reflect varying degrees of similarity between sensitive groups—tend to exhibit more severe fairness degradation. These results highlight the role of both demographic complexity and graph topology in shaping the vulnerabilities of GNNs to fairness attacks.

## 2 Scope of reproducibility

The objective of this study is to reproduce and validate the key findings presented in the original paper, *Are Your Models Still Fair? Fairness Attacks on Graph Neural Networks via Node Injections*, by Luo et al. (2024). The authors introduce NIFA, a gray-box poisoning attack designed to compromise the fairness of Graph Neural Networks (GNNs) while maintaining minimal impact on their utility. In this reproducibility study, we aim to confirm the following claims and explore their implications:

- **Claim 1**: NIFA can significantly degrade fairness in GNNs, measured by increased $\Delta_{SP}$ (Statistical Parity) and $\Delta_{EO}$ (Equal Opportunity), while maintaining negligible utility loss across diverse datasets and models, including fairness aware GNNs.

- **Claim 2**: NIFA achieves state-of-the-art performance with a perturbation rate as low as 1%, ensuring scalability and minimal detectability.

- **Claim 3**: Extensive experiments confirm that NIFA outperforms existing fairness and utility attack baselines across multiple datasets.

We reproduce the core experiments of the original NIFA paper to evaluate its primary claims across classical and fairness-aware GNN models. While Luo et al. (2024) benchmarked NIFA against six baseline attacks—three targeting utility and three targeting fairness—we were only able to reliably reproduce two: TDGIA (Zou et al., 2021) and FA-GNN (Hussain et al., 2022). The remaining baselines lacked accessible or compatible implementations, and integrating them would have required substantial reengineering across codebases.

In addition to this partial reproduction, we extend NIFA's evaluation in two directions to better reflect real-world fairness concerns:

- **Multi-Class Sensitive Attributes:** We successfully extend NIFA to handle datasets with multi-class sensitive attributes, reflecting more complex real-world scenarios. This will enable an evaluation of whether the attack can remain effective and scalable in settings with higher attribute diversity.

- **Impact of Homophily on NIFA:** We explore how varying homophily levels in graph structures affect NIFA's effectiveness and fairness, focusing on the interplay between homophily and bias propagation due to node injection. We aim to determine if certain structural properties make networks more susceptible to fairness degradation.

This study should be interpreted as a scoped reproduction and extension—not a comprehensive validation—of NIFA. Our findings offer targeted insights into its generalizability, but do not evaluate all claims or baselines from the original work.

## 3 Methodology

### 3.1 Problem Formulation

We adopt notation conventions from Luo et al. (2024). A graph $\mathcal{G} = (\mathcal{V}, \boldsymbol{A}, \boldsymbol{X})$ consists of nodes $\mathcal{V}$, an adjacency matrix $\boldsymbol{A} \in \mathbb{R}^{|\mathcal{V}| \times |\mathcal{V}|}$, and a feature matrix $\boldsymbol{X} \in \mathbb{R}^{|\mathcal{V}| \times D}$. Each node $v \in \mathcal{V}$ has a label $y_v \in \mathcal{Y}$ and a sensitive attribute $s$. While prior studies limit $s$ to binary values, we generalize $s$ to a $k$-class setting ($s \in 0, 1, ..., k-1$), where $k$ denotes the number of classes in a sensitive attribute. A GNN-based function $f_\theta$ with parameters $\theta$ is trained to predict labels using graph signals.

**Fairness-related Concepts.** In alignment with prior works (Dai & Wang, 2021; Dong et al., 2022; Ling et al., 2023), the authors focus on group fairness. Here, fairness is assessed across groups defined by sensitive attributes. The sensitive attribute divides the node set $\mathcal{V}$ into two disjoint groups: $\mathcal{V}_0$, where $s = 0$, and $\mathcal{V}_1$, where $s = 1$. Group fairness is assessed using two primary metrics: Statistical Parity (SP) and Equal Opportunity (EO). These metrics evaluate the independence of predictions and correct predictions, respectively, from the sensitive attribute.

**SP.** Statistical Parity requires that the model's predictions are independent of the sensitive attribute. Formally, for any predicted class $y \in \mathcal{Y}$, the following condition should hold:

$$P(\hat{y}_v = y | s = 0) = P(\hat{y}_v = y | s = 1) \tag{1}$$

where $\hat{y}_v$ denotes the predicted label for node $v$. Statistical Parity ensures that the distribution of predictions is uniform across sensitive groups, regardless of their occurrence in the graph.

**EO.** Equal Opportunity focuses on the model's ability to predict the correct label independently of the sensitive attribute. For any true class $y \in \mathcal{Y}$, it requires that:

$$P(\hat{y}_v = y | y_v = y, s = 0) = P(\hat{y}_v = y | y_v = y, s = 1) \tag{2}$$

Where $y_v$ represents the true label of node $v$. Equal Opportunity ensures that the likelihood of correct predictions is consistent across sensitive groups.

The degree to which fairness is violated is quantified using two metrics, $\Delta_{SP}$ and $\Delta_{EO}$, which measure deviations from the ideal conditions described above. These metrics are defined as follows:

$$\Delta SP = \mathbb{E}[|P(\hat{y} = y \mid s = 0) - P(\hat{y} = y \mid s = 1)|] \tag{3}$$

$$\Delta EO = \mathbb{E}[|P(\hat{y} = y \mid y_v, s = 0) - P(\hat{y} = y \mid y_v, s = 1)|] \tag{4}$$

For both metrics, smaller values indicate better fairness, with $\Delta_{SP} = 0$ and $\Delta_{EO} = 0$ representing perfect fairness.

**Attack Definition.** Following the line of previous attacks on GNNs (Sun et al., 2020; Zhang et al., 2024a), the authors launch a fairness-targeted attack on GNN models through the application of node injection during the training phase. Operating under a gray-box setting, the attacker has access to the graph structure $\mathcal{G}$, node labels $\mathcal{Y}$, and sensitive attributes $s$. However, the attacker lacks knowledge of the model architecture or parameters.

The attack involves injecting malicious nodes $\mathcal{V}_I$ into the graph during training to degrade fairness while maintaining model accuracy. The modified graph $\mathcal{G}' = (\mathcal{V}', \boldsymbol{A}', \boldsymbol{X}')$ includes injected nodes with trained features and connections. The goal is to maximize $\Delta_{SP}$ and $\Delta_{EO}$ under constraints on the number of injected nodes ($|\mathcal{V}_I| \leq b$) and their connectivity ($deg(v) \leq d$). The objective is expressed as:

$$\max_{\mathcal{G}'} \left| \mathcal{F}(f_{\theta^*}(\mathcal{V}, \mathcal{G}')) \right| \quad \text{s.t.} \quad \arg\max_{\theta^*} \mathcal{M}(f_{\theta^*}(\mathcal{V}, \mathcal{G}')), \quad |\mathcal{V}_I| \leq b, \quad deg(v)_{v \in \mathcal{V}_I} \leq d$$

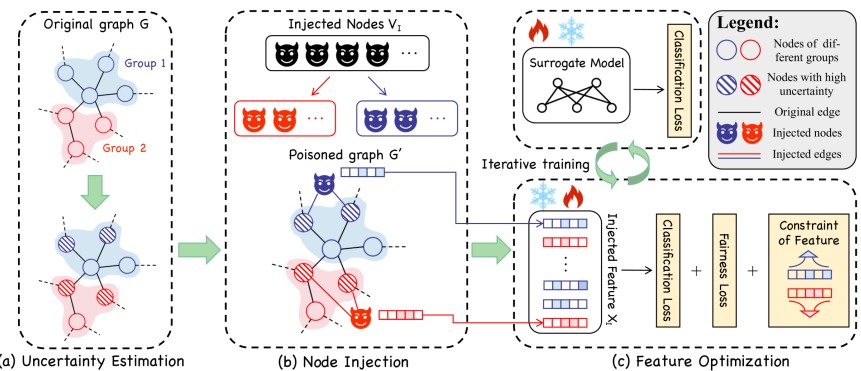

Figure 1: Overview of the NIFA framework (Luo et al., 2024)

where $\mathcal{F}(\cdot)$ and $\mathcal{M}(\cdot)$ are evaluation functions for fairness and utility, respectively. Constraints on the number of injected nodes ($|\mathcal{V}_I| \leq b$) and their connectivity ($deg(v)_{v \in \mathcal{V}_I} \leq d$) ensure that the attack remains unnoticeable and deceptive to the defenders, both $b$ and $d$ are predefined budgets.

## 3.2 Model framework

The NIFA framework combines node injection and feature optimization to degrade fairness in GNNs. It follows two principles for node injection: (1) Uncertainty-maximization, which uses a Bayesian GNN (Liu et al., 2022) to target nodes with high uncertainty near decision boundaries, and (2) Homophily-increase, which ensures injected nodes connect exclusively to nodes within the same sensitive group, amplifying intra-group information propagation and increasing the homophily ratio.

Injected nodes' features are optimized using multiple objectives to balance fairness degradation and utility preservation. An iterative strategy prevents overfitting to surrogate model parameters. The framework's components—uncertainty maximization, node injection, and feature optimization—are visualized in Figure 1 and detailed below.

**Uncertainty-maximization.** Nodes with high uncertainty are selected for injection as their predictions are more vulnerable to adversarial attacks. A Bayesian GNN with Monte Carlo (MC) dropout approximates parameter distributions (Liu et al., 2022), introducing variability in predictions through sampled dropout masks. Uncertainty scores are computed as the variance of predictions across samples:

$$U = \text{Var}_{i=1}^{T}\big(f_{\theta_{B_i}}(\mathcal{V}, \mathcal{G})\big).$$

where $T$ is the number of samples and $f_{\theta_{B_i}}(\cdot)$ is the mapping function with the $i$-th sampled Bayesian GNN parameters $\theta_{B_i}$. Nodes with the top $k\%$ uncertainty within each sensitive group are selected, with $k$ as a tunable hyperparameter.

**Homophily-increase.** Injected nodes are distributed among sensitive groups and connect to $d$ random target nodes within the same group, where $d$ is a hyperparameter (Figure 1b). This strategy prevents intra-group information propagation, exacerbating unfairness by increasing node-level homophily:

$$\mathcal{H}_u = \frac{\sum_{v \in \mathcal{N}_u} \mathbb{1}(s_u = s_v)}{|\mathcal{N}_u|},$$

where $\mathcal{N}_u$ represents the neighbors of node $u$, $s_u$ is the sensitive attribute of $u$, and $\mathbb{1}(\cdot)$ is the indicator function.

**Feature optimization.** Injected nodes' features $\mathbf{X}_i$ are optimized to enhance the effectiveness of NIFA. In a gray-box attack setting, attackers lack direct visibility of victim models and instead train a surrogate

GNN model $\mathcal{S}$ on the poisoned graph $\mathcal{G}'$. The surrogate model $\mathcal{S}$ is trained iteratively alongside $\mathbf{X}_i$ using different objectives, preventing overfitting to specific model parameters. While $\mathcal{S}$ follows a standard GNN training process with cross-entropy loss, the optimization of $\mathbf{X}_i$ incorporates multiple objective functions to maximize unfairness while maintaining utility. Specifically, the optimization framework balances three key components: (1) classification loss to preserve predictive performance, (2) fairness loss to amplify disparities, and (3) a feature constraint to reinforce information differences across sensitive groups. The details of these loss functions and their implementation are presented in Appendix A.

### 3.3 Additional Experiments

The previous sections have outlined the foundational work presented in (Luo et al., 2024). In this section, we provide a detailed explanation of the contributions of this study. As in the original NIFA paper, we focus exclusively on node classification tasks, meaning edge classification is not considered in this study.

**Multi-class sensitive attributes.** In real-world scenarios, sensitive attributes often tend to be non-binary (Duong & Conrad, 2023). To adapt the NIFA attack to these complex scenarios, the attack is generalized to operate on multi-class sensitive attributes. This extension involves two main components. Firstly, the fairness metrics must be generalized to accommodate multiple sensitive attribute classes. Secondly, the attack methodology requires adaptation to effectively address multiple sensitive groups.

To maintain consistency with the binary-sensitive attribute metrics established in the original paper, the One-vs-All (OvA) and One-vs-One (OvO) approaches are explored. These methods, commonly used in multi-class classification, allow multi-class problems to be approached by aggregating the results of binary classifiers. In the same way, these techniques can be used to generalize the binary metrics to multi-class sensitive attributes by aggregating binary metrics. Comparable approaches have been utilized in prior research, where the One-vs-All method has been successfully applied to extend fairness metrics (Rouzot et al., 2022; Krasanakis et al., 2018).

In the One-vs-All approach, the binary metrics, as introduced in Equations (1 & 2), are computed for each sensitive group against the remainder of the nodes. These binary results are then aggregated into a single metric by computing the expected value over all sensitive groups $a \in \mathcal{A}$ and all labels $y \in \mathcal{Y}$:

$$\Delta_{SP\text{-}OvA} = \mathbb{E}[|P(\hat{y} = y|s = a) - P(\hat{y} = y|s \neq a)|] \tag{5}$$

$$\Delta_{EO\text{-}OvA} = \mathbb{E}[|P(\hat{y} = y|y_v = y, s = a) - P(\hat{y} = y|y_v = y, s \neq a)|] \tag{6}$$

In contrast, the One-vs-One approach computes the expected value for the results of the binary metrics, which are computed for all possible combinations of the sensitive groups $(a_1, a_2) \in \{(x, y)|x, y \in \mathcal{A}, x \neq y, x < y\}$ for all labels $y \in \mathcal{Y}$:

$$\Delta_{SP\text{-}OvO} = \mathbb{E}[|P(\hat{y} = y|s = a_1) - P(\hat{y} = y|s = a_2)|] \tag{7}$$

$$\Delta_{EO\text{-}OvO} = \mathbb{E}[|P(\hat{y} = y|y_v = y, s = a_1) - P(\hat{y} = y|y_v = y, s = a_2)|] \tag{8}$$

Two modifications to the model framework are applied to ensure the attack methodology works. First, the node injection process must be adjusted to ensure an equal proportion of injected nodes connect to each sensitive group while maintaining the homophily-increase principle. Second, the most significant change lies in feature optimization, where generalized metrics must be transformed into corresponding loss functions. The existing loss functions, detailed in Appendix A, are extended in line with the generalization of metrics. For the One-vs-All approach, $L_{SP}$ and $L_{EO}$ are computed for each sensitive group versus the remaining groups, and the mean of these losses is used for optimization. Whereas in the One-vs-One approach, the mean losses are calculated across all pairwise combinations of sensitive attributes. The resulting mean losses will replace the $L_{SP}$ and $L_{EO}$ in the optimization objective.

**Homophily Evaluation.** To investigate the relationship between homophily and the effectiveness of NIFA, we adopt a systematic methodology involving graph generation, GNN training, and fairness analysis. Graph structures with varying homophily levels are constructed using a synthetic dataset. For these graphs, the

Directed Preferential Attachment with Homophily (DPAH) model (Espín-Noboa et al., 2022) is employed to precisely control homophily parameters, enabling exploration across a spectrum of heterophilic to homophilic configurations. To ensure a fair comparison, we also calculate the overall homophily rates for both the synthetic and original datasets. These configurations are shown in Figure 2. Beyond these predefined graphs, we explore a wide range of homophily rate combinations, detailed in Appendix D, building on insights from Ferrara et al. (2022).

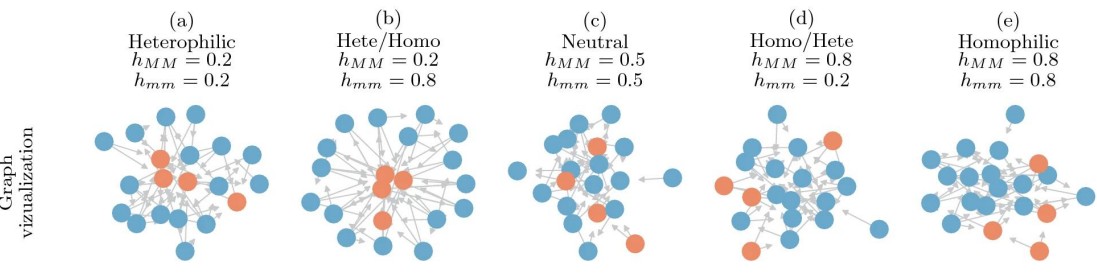

Figure 2: Graph examples of different levels of homophily (Espín-Noboa et al., 2022)

The standard GCN model used in Luo et al. (2024) is used to train generated graphs under consistent experimental conditions. Similarly to the original paper, performance is evaluated on their ability to predict node labels. The NIFA framework is then applied across all homophily scenarios, with node injections optimized to target fairness. The perturbation rate is maintained at 1% to ensure minimal detectability, as per the original study. The impact of the attack is assessed by measuring changes in fairness metrics ($\Delta_{SP}$ and $\Delta_{EO}$) and utility preservation.

Fairness impact is further analyzed by examining how varying homophily levels influence NIFA's ability to isolate minority nodes and exacerbate fairness disparities. The attack's effectiveness is quantified by evaluating the degradation in both fairness and model utility, with attention to the role of structural graph properties in mediating these effects. Through this comprehensive framework, we aim to provide insights into the interplay between homophily and fairness vulnerabilities, as well as strategies for mitigating adversarial attacks in graph-based models.

## 4 Experimental Setup and Code

The code to run NIFA on the four classic GNN models, including datasets and package requirements, is publicly available on the authors' official code repository[1]. This code runs without requiring major modifications, although installing the correct packages was necessary as some library-specific updates have been made since. Code for reproducing the other experiments, such as running fairness-aware models and performing utility and fairness attacks, was not provided. To perform these experiments, their respective code repositories were independently accessed and made compatible with the NIFA-generated graphs. An updated list of requirements was created to ensure compatibility with our hardware, which is also included in the code repository. Each reproduction experiment is conducted five times, following the methodology outlined in the original paper.

### 4.1 Datasets

**Original Datasets.** The experiments conducted in the original paper utilized three different real-world datasets: **Pokec-z**, **Pokec-n**[2], and **DBLP**[3]. Pokec-z and Pokec-n are subsets of the Pokec network. Accord-

---

[1]https://github.com/CGCL-codes/NIFA
[2]https://snap.stanford.edu/data/soc-pokec.html
[3]DBLP. Computer Science Bibliography

ing to Takac & Zabovsky (2012), Pokec is the most popular social network in Slovakia and is also widely used in the Czech Republic. In this dataset, nodes represent users, while edges represent unidirectional following relationships (Luo et al., 2024).

The DBLP dataset is a citation dataset containing a digital library of comprehensive coverage of database literature (Elmacioglu & Lee, 2005). Each node represents an author, and two authors are connected if they have co-authored at least one paper. The node features are derived from words extracted from the published papers of the corresponding author (Luo et al., 2024). The specifics of each dataset, as well as their respective homophily rates, are summarized in Table 4 in Appendix C. Pokec-z and Pokec-n contain significantly larger numbers of nodes and edges than DBLP, while DBLP has the highest feature dimensionality.

**Multi-class Sensitive Attribute Dataset.** To the best of our knowledge, there are no graph datasets for node classification with explicit multi-class sensitive attributes. To accommodate for this, the method proposed by Purificato et al. (2024) is adopted, which involves augmenting a dataset that has binary sensitive attributes to produce multi-class sensitive attributes. This augmentation is achieved by converting other potentially sensitive features into classes, for example, through binning. We follow the approach introduced by Purificato et al. (2024), who applied this technique to the Pokec dataset by creating a new sensitive attribute. Specifically, the age feature was binned into the following ranges to form five age groups: 0-17, 18-23, 24-28, 29-35, and 36 and above. These experiments aim to compare the OvA and OvO approaches to generalizing the NIFA attack to multi-class sensitive attributes.

**Synthetic Dataset.** To investigate the impact of homophily rates, we construct a synthetic dataset based on the work by Espín-Noboa et al. (2022). The dataset consists of synthetic networks generated using the Directed Preferential Attachment with Homophily (DPAH) model[4], which integrates both preferential attachment and homophily. Further details on the model's structure are provided in Appendix D.

The dataset lacks explicit node features and only includes binary labels for majority and minority groups. To adapt this dataset for use within the NIFA pipeline, we generate proxy features by introducing a synthetic node classification task. We first assign proxy class labels to the nodes based on two Gaussian distributions. For a given node $i$, its proxy class label $y_i$ is determined by sampling from two Gaussian distributions $\mathcal{N}(\mu_1, \sigma^2)$ and $\mathcal{N}(\mu_2, \sigma^2)$, where the mean vectors $\mu_1$ and $\mu_2$ represent the centers of the proxy classes, and $\sigma^2$ controls the variance within each class. These proxy labels serve as a basis for creating features required for the NIFA pipeline.

To construct the feature space, a $n$-dimensional feature vector $\mathbf{x}_i \in \mathbb{R}^n$ was generated for each node $i$, where each feature dimension is sampled from the Gaussian corresponding to its assigned proxy label. Specifically:

$$\mathbf{x}_i \sim \begin{cases} \mathcal{N}(\mu_1, \sigma^2), & \text{if } y_i = 0, \\ \mathcal{N}(\mu_2, \sigma^2), & \text{if } y_i = 1. \end{cases}$$

A correlation parameter $\rho$ is introduced to establish a statistical relationship between the Gaussian distributions and the proxy labels. This ensures that the feature space effectively reflects the proxy labels, making the nodes distinguishable and enabling the creation of a meaningful node classification task. The sensitive attribute $s_i \in \{0, 1\}$, indicating whether a node belongs to the majority ($s_i = 0$) or minority ($s_i = 1$) group, remains binary. We divide the dataset into train (70%), validation (15%), and test (15%) splits. Each split preserves the original distribution of sensitive attributes and proxy labels to minimize sampling bias.

## 4.2 Model descriptions

**Classical GNN Models.** To reproduce the original NIFA results on classical models, GCN (Kipf & Welling, 2017), GraphSAGE (Hamilton et al., 2017), APPNP (Gasteiger et al., 2022), and SGC (Wu et al., 2019), the code provided on the GitHub repository was used without significant modifications. Further clarifications on these models are found in Appendix B.1.

---

[4]https://github.com/gesiscss/Homophilic_Directed_ScaleFree_Networks

**Fairness Aware Models.** Reproducing the results of the fairness aware models, specifically FairGNN (Dai & Wang, 2021), FairVGNN (Wang et al., 2022), and FairSIN (Ju et al., 2023) (see Appendix B.1 for more details), proved to be more complex than reproducing those of the classical models. Adjustments to the code of the fairness-aware models were necessary to ensure valid results. Specifically, NIFA's poisoned graphs needed to be stored for each dataset and loaded through a custom data loading function to maintain compatibility with the fairness model pipelines. Additionally, minor modifications to the training logic, including adjustments to the loss functions, were required. Finally, the exact fairness metrics outlined in subsection 3.1 had to be implemented within each fairness model.

**Baseline Attack Models.** The authors of the original paper compared NIFA to various competitors, including attacks based on utility: AFGSM (Wang et al., 2020), TDGIA (Zou et al., 2021), $G^2$A2C (Ju et al., 2023), and attacks based on fairness: FA-GNN (Hussain et al., 2022), FATE (Kang et al., 2023), G-FairAttack (Zhang et al., 2024b). Due to time limitations and the need to reproduce these models independently, two of these models were reproduced: the utility-based attack TDGIA and the fairness-based attack FA-GNN. Their details are found in Appendix B.2.

To reproduce the FA-GNN results, two minor modifications were made to the code: the graph-loading process was adjusted, and the fairness metrics detailed in subsection 3.1 were implemented. To obtain results for TDGIA, the graphs were poisoned by TDGIA and evaluated on the NIFA fairness metrics. Three modifications were made to the codebase: the TDGIA graph-handling method was adjusted, a graph transformation step was implemented in the NIFA code, and parts of the NIFA implementation were disabled to prevent already poisoned graphs from being reattacked.

### 4.3 Computational requirements

All our experiments were conducted utilizing NVIDIA A100 GPUs with CUDA 12.1.1. We tried to closely replicate the Python environment specified by the authors to ensure consistency. Specifically, we used Python 3.10.16 and PyTorch 2.4.0. For the Deep Graph Library (DGL), we employed version 2.4.0, which requires PyTorch $\leq$ 2.4.0, a constraint we adhered to for compatibility. Additionally, we incorporated the PyTorch Geometric (PyG) library version 2.6.1, along with auxiliary libraries like torch- scatter and torch-sparse, all compatible with CUDA 12.1.1.

## 5 Results

### 5.1 Results reproducing the original paper

The reproduced outcomes of the models described in subsection 4.2 are presented in Table 1. The reproduced results for utility- and fairness-based attacks are presented in Table 2. In alignment with the original work, we report the mean and standard deviation for all results in this section. The results from the original paper are provided in Appendix E.

The original paper makes three key claims about the effectiveness of NIFA: (1) it significantly degrades fairness (measured via $\Delta_{SP}$ and $\Delta_{EO}$), (2) it does so without compromising utility, and (3) it outperforms existing fairness and utility attacks across datasets.

Claims 1 and 2 are primarily supported by substantial increases in $\Delta_{SP}$ and $\Delta_{EO}$ across classical GNN models, while maintaining high classification accuracy. For example, in the original results on the Pokec-z dataset, the $\Delta_{SP}$ and $\Delta_{EO}$ values for GCN increased from 7.13% and 5.10% to 17.36% and 15.59%, respectively. Even on fairness-aware models, the impact was reported as significant—for instance, FairVGNN's $\Delta_{EO}$ on Pokec-z rose from 2.59% to 9.28%, a nearly fourfold increase. Meanwhile, utility metrics across models dropped only slightly, usually by a few percentage points.

In our reproduction, these claims are partially validated. For classical GNN models, our results closely mirror the original findings. While the absolute values of $\Delta_{SP}$ and $\Delta_{EO}$ are slightly lower in our experiments, the relative increase from pre- to post-attack remains consistent. On Pokec-z, for example, all classical models show an approximate twofold increase in both fairness metrics following the attack. Similar trends are

Table 1: Attack performance of NIFA on different victim GNN models. Results are reported in percentages (%). To showcase trends similar to the original paper, we **bold** the results where NIFA successfully degraded the fairness of victim GNN models.

| | | Pokec-z | | | Pokec-n | | | DBLP | | |
|---|---|---|---|---|---|---|---|---|---|---|
| | | Accuracy | $\Delta_{SP}$ | $\Delta_{EO}$ | Accuracy | $\Delta_{SP}$ | $\Delta_{EO}$ | Accuracy | $\Delta_{SP}$ | $\Delta_{EO}$ |
| **GCN** | Before | $71.27 \pm 0.22$ | $7.74 \pm 0.39$ | $5.82 \pm 0.34$ | $71.12 \pm 0.34$ | $0.74 \pm 0.47$ | $2.08 \pm 0.84$ | $96.13 \pm 0.76$ | $3.74 \pm 1.08$ | $2.31 \pm 1.32$ |
| | After | $70.99 \pm 0.55$ | $\mathbf{15.74 \pm 1.03}$ | $\mathbf{13.91 \pm 1.07}$ | $70.38 \pm 0.48$ | $\mathbf{7.94 \pm 0.71}$ | $\mathbf{7.89 \pm 0.91}$ | $92.96 \pm 1.64$ | $\mathbf{11.12 \pm 4.37}$ | $\mathbf{18.43 \pm 4.10}$ |
| **GraphSAGE** | Before | $70.58 \pm 0.76$ | $5.01 \pm 1.48$ | $3.50 \pm 0.77$ | $68.77 \pm 0.34$ | $1.65 \pm 1.31$ | $2.34 \pm 1.04$ | $96.03 \pm 0.37$ | $4.65 \pm 2.81$ | $4.40 \pm 0.65$ |
| | After | $69.62 \pm 0.56$ | $\mathbf{5.81 \pm 1.19}$ | $\mathbf{4.08 \pm 0.89}$ | $68.93 \pm 1.19$ | $\mathbf{3.32 \pm 1.88}$ | $\mathbf{3.56 \pm 1.91}$ | $94.42 \pm 0.92$ | $\mathbf{8.08 \pm 1.24}$ | $\mathbf{14.08 \pm 1.90}$ |
| **APPNP** | Before | $69.28 \pm 0.83$ | $7.05 \pm 0.50$ | $5.29 \pm 0.49$ | $68.573 \pm 0.37$ | $5.02 \pm 0.75$ | $5.34 \pm 0.74$ | $96.53 \pm 0.33$ | $4.39 \pm 0.88$ | $2.94 \pm 1.21$ |
| | After | $69.46 \pm 0.43$ | $\mathbf{18.88 \pm 1.29}$ | $\mathbf{17.42 \pm 1.33}$ | $68.32 \pm 0.46$ | $\mathbf{7.74 \pm 2.61}$ | $\mathbf{7.40 \pm 2.58}$ | $92.26 \pm 1.05$ | $\mathbf{12.61 \pm 4.88}$ | $\mathbf{18.94 \pm 6.33}$ |
| **SGC** | Before | $66.65 \pm 3.42$ | $6.63 \pm 1.63$ | $5.38 \pm 0.95$ | $66.42 \pm 3.25$ | $4.18 \pm 2.23$ | $4.69 \pm 1.93$ | $96.58 \pm 0.47$ | $4.07 \pm 0.47$ | $2.67 \pm 0.68$ |
| | After | $67.02 \pm 1.56$ | $\mathbf{14.53 \pm 2.75}$ | $\mathbf{12.99 \pm 2.74}$ | $67.43 \pm 0.69$ | $\mathbf{9.93 \pm 3.00}$ | $\mathbf{9.58 \pm 2.96}$ | $92.81 \pm 0.76$ | $\mathbf{10.21 \pm 5.55}$ | $\mathbf{16.44 \pm 7.05}$ |
| **FairGNN** | Before | $72.10 \pm 0.12$ | $5.12 \pm 0.58$ | $3.04 \pm 0.63$ | $70.87 \pm 0.21$ | $0.91 \pm 0.61$ | $1.39 \pm 0.46$ | $95.38 \pm 0.41$ | $3.27 \pm 2.28$ | $3.79 \pm 1.16$ |
| | After | $70.98 \pm 0.14$ | $\mathbf{5.30 \pm 0.64}$ | $\mathbf{3.17 \pm 0.67}$ | $71.07 \pm 0.28$ | $\mathbf{2.02 \pm 1.42}$ | $\mathbf{2.86 \pm 0.90}$ | $95.53 \pm 0.29$ | $\mathbf{4.46 \pm 1.53}$ | $\mathbf{8.54 \pm 2.12}$ |
| **FairVGNN** | Before | $66.65 \pm 0.91$ | $1.35 \pm 0.39$ | $3.48 \pm 0.51$ | $64.75 \pm 2.81$ | $1.26 \pm 0.66$ | $1.31 \pm 0.70$ | $95.58 \pm 0.47$ | $3.97 \pm 0.98$ | $1.91 \pm 0.68$ |
| | After | $66.43 \pm 0.95$ | $\mathbf{1.95 \pm 1.16}$ | $\mathbf{3.69 \pm 1.26}$ | $65.67 \pm 2.84$ | $\mathbf{2.89 \pm 2.12}$ | $\mathbf{3.55 \pm 1.32}$ | $90.25 \pm 1.12$ | $\mathbf{11.13 \pm 5.75}$ | $\mathbf{17.70 \pm 7.91}$ |
| **FairSIN** | Before | $68.3 \pm 0.35$ | $3.79 \pm 1.85$ | $3.96 \pm 1.67$ | $61.34 \pm 3.82$ | $3.33 \pm 1.67$ | $5.00 \pm 1.87$ | $94.32 \pm 0.67$ | $1.22 \pm 0.78$ | $3.40 \pm 1.48$ |
| | After | $64.14 \pm 5.66$ | $\mathbf{4.33 \pm 2.81}$ | $3.65 \pm 2.58$ | $64.51 \pm 2.7$ | $1.80 \pm 0.95$ | $4.14 \pm 1.11$ | $90.50 \pm 1.40$ | $\mathbf{13.47 \pm 3.09}$ | $\mathbf{37.31 \pm 5.74}$ |

observed across the other two datasets. Moreover, utility degradation is consistently minor, supporting the claim that NIFA is effective without significantly impacting model performance.

However, for fairness-aware GNN models, our results diverge from those reported by Luo et al. (2024). In the original study, NIFA causes large increases in $\Delta_{SP}$ and $\Delta_{EO}$ on the Pokec-z and Pokec-n datasets—ranging from twofold to over twentyfold changes. In contrast, our reproduction shows only modest increases on these datasets, with fairness metrics even decreasing in three of the evaluated cases. These discrepancies may stem from implementation differences or incompatibilities across fairness-aware model codebases, as discussed in Section 6.1. Based on these results, we conclude that Claims 1 and 2 hold for classical models, but only partially hold for fairness-aware models.

Finally, while the original study also emphasizes NIFA's stealth—by restricting node injection to just 1% of the graph—we did not directly evaluate detectability (e.g., via anomaly detection methods). We adhere to the original perturbation budget, but consider the detectability aspect of Claim 2 outside the scope of our study.

Table 2: Accuracy and Fairness performance of attack launched by the different attackers. The results are reported in percentages (%).

| | Pokec-z | | | Pokec-n | | | DBLP | | |
|---|---|---|---|---|---|---|---|---|---|
| | Accuracy | $\Delta_{SP}$ | $\Delta_{EO}$ | Accuracy | $\Delta_{SP}$ | $\Delta_{EO}$ | Accuracy | $\Delta_{SP}$ | $\Delta_{EO}$ |
| **Clean** | $71.27 \pm 0.22$ | $7.74 \pm 0.39$ | $5.82 \pm 0.34$ | $71.12 \pm 0.34$ | $0.74 \pm 0.47$ | $2.08 \pm 0.84$ | $95.88 \pm 1.61$ | $3.84 \pm 0.34$ | $1.91 \pm 0.75$ |
| *Utility attacks on GNNs* | | | | | | | | | |
| **TDGIA** | $66.56 \pm 2.65$ | $9.60 \pm 1.47$ | $7.92 \pm 1.59$ | $67.09 \pm 1.94$ | $0.64 \pm 0.37$ | $1.31 \pm 0.29$ | $90.90 \pm 0.58$ | $5.97 \pm 0.76$ | $3.61 \pm 0.64$ |
| *Fairness attacks on GNNs* | | | | | | | | | |
| **FA-GNN** | $71.07 \pm 0.76$ | $5.89 \pm 2.23$ | $4.41 \pm 1.98$ | $71.11 \pm 0.51$ | $3.21 \pm 0.62$ | $2.57 \pm 1.61$ | $95.39 \pm 0.90$ | $1.63 \pm 1.20$ | $3.15 \pm 1.48$ |
| **NIFA** | $70.99 \pm 0.55$ | $\mathbf{15.74 \pm 1.03}$ | $\mathbf{13.91 \pm 1.07}$ | $70.38 \pm 0.48$ | $\mathbf{7.94 \pm 0.71}$ | $\mathbf{7.89 \pm 0.91}$ | $92.96 \pm 1.64$ | $\mathbf{11.12 \pm 4.37}$ | $\mathbf{18.43 \pm 4.10}$ |

Claim 3, which asserts that NIFA outperforms existing fairness and utility attacks, is supported in the original paper through comparisons with six baselines—three targeting utility (e.g., TDGIA) and three targeting fairness (e.g., FA-GNN).

To assess this claim, we reproduced one attack from each category: TDGIA and FA-GNN. Across all three datasets, NIFA consistently resulted in higher fairness degradation than either baseline. For example, on the Pokec-n dataset, the post-attack $\Delta SP$ scores were 0.64% for TDGIA, 3.21% for FA-GNN, and 7.94% for NIFA. These results align with the original paper's reported trends, suggesting that NIFA remains a strong adversary under fairness metrics.

However, we do not evaluate the full set of baselines due to the absence of accessible or stable implementations for several attacks. Integrating these into the NIFA pipeline would have required extensive adaptation and might not have reflected the original experimental setup. Moreover, we observed discrepancies in absolute

metric values across all methods, which may be attributed to differences in codebases, evaluation pipelines, or random initialization. As such, we interpret our findings as partial support for the third claim, rather than a comprehensive benchmark.

While our results confirm that NIFA maintains model utility post-attack—consistent with Claim 2—we did not explicitly assess its detectability. The original paper emphasizes NIFA's stealth by limiting perturbation to 1% of the graph. We follow this constraint, but do not evaluate detectability via graph anomaly detection or visibility metrics. We therefore consider the stealth-related component of Claim 2 outside the scope of this reproduction.

### 5.2 Results beyond the original paper

We conducted two additional experiments to evaluate NIFA's attack on fairness: one extending the sensitive attribute to a multi-class setting and another analyzing its effects across different graph structures.

**Multi-class sensitive attributes.** The experimental results of NIFA on the augmented Pokec datasets with multi-class sensitive attributes described in section 4.1 are presented in Table 3. These results include both the OvA and OvO approaches, where the mean and standard deviation across five random seeds are reported. Note that the fairness metrics correspond to the respective OvA and OvO versions.

Table 3: NIFA attack performance on augmented Pokec-z and Pokec-n datasets using GCN. Results are in percentages (%) with the best fairness metrics in **bold**.

| | Pokec-z | | | Pokec-n | | |
|---|---|---|---|---|---|---|
| | Accuracy | $\Delta_{SP}$ | $\Delta_{EO}$ | Accuracy | $\Delta_{SP}$ | $\Delta_{EO}$ |
| *OvA* | | | | | | |
| **Before** | $71.11 \pm 0.14$ | $39.19 \pm 0.38$ | $25.58 \pm 0.32$ | $70.64 \pm 0.22$ | $40.81 \pm 1.08$ | $26.51 \pm 3.12$ |
| **After** | $70.88 \pm 0.63$ | $\mathbf{42.84 \pm 0.46}$ | $\mathbf{38.89 \pm 0.76}$ | $71.12 \pm 0.70$ | $\mathbf{46.25 \pm 0.82}$ | $\mathbf{34.52 \pm 1.37}$ |
| *OvO* | | | | | | |
| **Before** | $71.31 \pm 0.17$ | $37.74 \pm 0.39$ | $31.03 \pm 0.37$ | $70.67 \pm 0.17$ | $40.17 \pm 0.85$ | $26.05 \pm 2.86$ |
| **After** | $71.29 \pm 0.49$ | $\mathbf{39.88 \pm 0.32}$ | $\mathbf{39.88 \pm 1.15}$ | $70.67 \pm 0.43$ | $\mathbf{43.98 \pm 0.33}$ | $\mathbf{33.36 \pm 0.34}$ |

The results of NIFA utilizing multi-class sensitive attributes indicate that the authors' claims remain valid in this setting. As observed in Table 3, the model's utility remains comparable to that of the model trained on the clean graph. Simultaneously, the fairness scores demonstrate that the attack was effective, further supporting the authors' claims. However, the absolute values are higher compared to the original results, which we attribute to the use of a different sensitive attribute that changes the feature set of the nodes.

**Impact of graph structures.** The impact of the NIFA attack on fairness under varying synthetic graph structures and homophily levels, as introduced in section 3.3, is presented in Figure 3. Results are reported as the mean and standard deviation across five random seeds. We focus on a minority group fraction of $f_m = 0.3$, which most closely reflects the demographic ratios in the original real-world datasets (see subsection 4.1 and Appendix C).

We find that fairness degradation due to NIFA is most pronounced in the heterophilic-homophilic configuration (Hete/Homo = 0.36). One possible explanation is that this structure introduces conflicting clustering tendencies—inter-group mixing in the heterophilic portion and tightly bound intra-group communities in the homophilic portion—allowing injected nodes to exploit these dynamics and disrupt group fairness more effectively. However, we also observe high variance in this regime across random seeds, suggesting that structural sensitivity and initialization effects play a non-negligible role.

Overall, the $\Delta_{SP}$ and $\Delta_{EO}$ metrics remain relatively stable across most homophily settings, with two exceptions: heterophilic-homophilic mixtures and highly homophilic graphs. The latter in particular exhibits greater variability in fairness degradation, potentially due to the increased density and redundancy in intra-group links, which may amplify stochastic effects in the attack's impact. These findings are consistent with

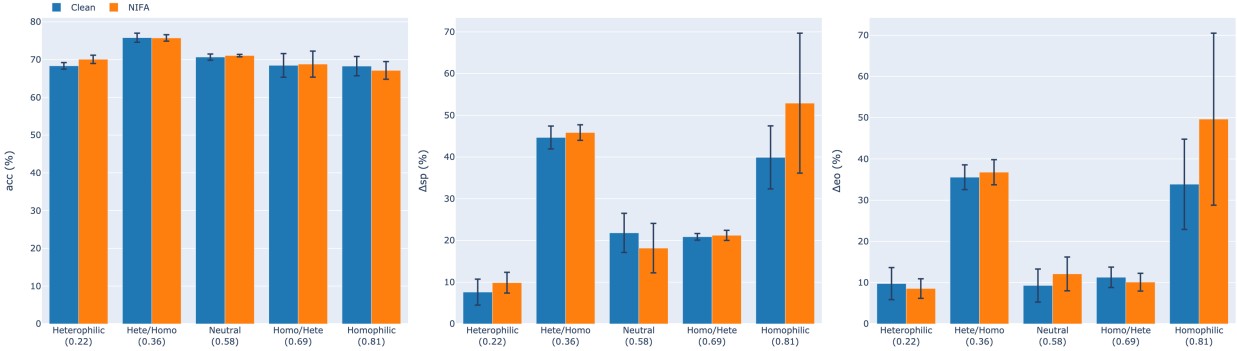

Figure 3: Effect of edge homophily on fairness metrics before and after the NIFA attack ($f_m = 0.3$).

NIFA's behavior on real datasets and highlight the need for further study of how structural graph properties modulate fairness vulnerability.

# 6 Discussion

## 6.1 Discussion of the results

The primary aim of this study was to reproduce and critically evaluate the three main claims presented by Luo et al. (2024) in *Are Your Models Still Fair? Fairness Attacks on Graph Neural Networks via Node Injections.* As discussed in subsection 5.1, we find partial support for Claims 1 and 2, while Claim 3 remains only partially supported.

Several factors likely contributed to discrepancies between our results and those reported in the original paper. First, only the classical GNN victim models were available in the official NIFA code repository. The fairness-aware models and baseline attacks had to be independently sourced and implemented, potentially introducing variations in architectural choices, preprocessing steps, or hyperparameters. Although we followed the reference implementations closely, subtle differences may explain the divergence in absolute performance metrics. Second, the original results may have been influenced by undocumented random seeds and environmental configurations, further complicating exact replication.

Our extension of NIFA to multi-class sensitive attributes produces fairness degradation patterns consistent with those observed in the binary case. This suggests that the NIFA attack generalizes to more complex demographic settings and supports the use of extended fairness metrics beyond binary attribute distinctions.

Moreover, our analysis of homophily indicates that graph structure significantly influences the success of fairness attacks. Graphs with mixed or high homophily exhibited greater fairness degradation, possibly due to denser intra-group connections that amplify bias propagation via injected nodes. These effects were most pronounced in synthetic graphs with heterophilic-homophilic structures, which also showed higher metric variability. These findings raise important questions about the structural factors influencing fairness vulnerabilities. Real-world datasets in the original NIFA study also exhibit high homophily, which may partially explain NIFA's effectiveness in those settings. However, due to limited data and high seed variance in homophilic settings, we refrain from drawing strong conclusions at this stage. A more systematic investigation of structural effects is left for future work.

Although NIFA is an adversarial attack method, understanding its behavior across varied structural and demographic conditions can inform the development of more robust, fair, and accountable GNN models. Fairness attacks expose structural weaknesses in deployed systems, particularly in socially sensitive domains such as hiring, lending, and healthcare. By extending evaluation to multi-class settings and diverse graph structures, our work contributes to the ongoing effort to identify and mitigate fairness vulnerabilities in graph-based machine learning.

Given the limited range of baseline attacks assessed, we emphasize that this study does not constitute a comprehensive validation of NIFA's claimed superiority. While we confirm key trends and broaden the evaluation scope, our findings should be interpreted as preliminary. Future work should incorporate additional baselines to more thoroughly assess fairness attack performance across settings.

### 6.2   Reflection: What was easy? What was difficult?

Once initial dependency issues were resolved, reproducing the attack performance of NIFA on the classical GNN models was relatively straightforward. The official GitHub repository was well-organized, and the README provided clear guidance for setting up and running these experiments.

However, several challenges arose during reproduction. The original codebase required CUDA versions older than 11.6.0 and included deprecated dependencies, particularly in relation to DGL. As a result, we were unable to fully adhere to the stated environment specifications and had to modify portions of the code to ensure compatibility with updated libraries.

Further difficulties emerged when reproducing results for fairness-aware models and baselines not included in the NIFA repository. The original paper referenced additional code repositories but did not specify how to integrate them with the NIFA attack pipeline. While NIFA's codebase was well-structured, the external repositories posed challenges, including adapting data-loading procedures for Pokec and DBLP, and modifying utility functions to align with the poisoned graph setup.

A detailed account of these implementation adjustments is provided in subsection 4.2.

### 6.3   Communication with original authors

We contacted the original authors of the paper twice to seek clarification on aspects of their implementation of the fairness-aware models and baseline attack models. The inquiries focused on understanding implementation details, such as confirming the configuration of key parameters, verifying any updates to the official code repository, and clarifying certain techniques used in experiments that were not fully documented in the paper. Both inquiries received prompt and helpful responses, for which their clarifications and timely assistance are greatly appreciated.

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

## Appendices

## A    Feature optimization

The feature optimization background is directly adapted from Luo et al. (2024).

**Classification loss.**  To ensure the utility of the victim model remains unaffected and avoid detection, the authors employ the cross-entropy loss as the first objective function:

$$L_{CE} = -\frac{1}{|\mathcal{V}^{tr}|} \sum_{u \in \mathcal{V}^{tr}} y_u \log h_u$$

where $\mathcal{V}^{tr}$ denotes the original training node set, $y_u$ is the true label of node $u$, and $h_u$ is the output logits of node $u$.

**Fairness loss.**  To maximize unfairness, two fairness loss functions are defined based on the metrics $\Delta_{SP}$ and $\Delta_O$:

$$L_{SP} = -\|\frac{1}{|\mathcal{V}_0^{tr}|} \sum_{u \in \mathcal{V}_0^{tr}} h_u - \frac{1}{|\mathcal{V}_1^{tr}|} \sum_{u \in \mathcal{V}_1^{tr}} h_u\|_2^2,$$

$$L_{EO} = -\|\sum_{y \in \mathcal{Y}} (\frac{1}{|\mathcal{V}_{0,y}^{tr}|} \sum_{u \in \mathcal{V}_{0,y}^{tr}} h_{u,y} - \frac{1}{|\mathcal{V}_{1,y}^{tr}|} \sum_{u \in \mathcal{V}_{1,y}^{tr}} h_{u,y})\|_2^2,$$

where $h_u \in \mathbb{R}^{|\mathcal{Y}|}$ represents the raw output for node $u$, and $h_{u,y}$ is the output for class $y$. $\mathcal{V}_{i,y}^{tr}$ denotes the set of training nodes with sensitive attribute $i$ and label $y$.

**Constraint of feature.**  To distinguish information introduced by injected nodes between sensitive groups during message propagation, the authors define a feature constraint:

$$L_{CF} = -\|\frac{1}{|\mathcal{V}_{I,0}|} \sum_{u \in \mathcal{V}_{I,0}} \mathbf{X}_{I,u} - \frac{1}{|\mathcal{V}_{I,1}|} \sum_{u \in \mathcal{V}_{I,1}} \mathbf{X}_{I,u}\|_2^2,$$

where $V_{I,i}$ is the set of injected nodes connected to sensitive group $i$.

**Overall loss.**  The overall loss for optimizing the injected nodes' features is defined as:

$$L = L_{CE} + \alpha \cdot L_{CF} + \beta \cdot (L_{SP} + L_{EO}),$$

where $\alpha$ and $\beta$ are hyperparameters controlling the weight of each term.

## B    Additional model descriptions

### B.1    Victim models

**GCN (Kipf & Welling, 2017):**  A GNN is based on convolutions that operate directly on graphs. All local node information is aggregated to generate graph representations, enabling semi-supervised learning on graph-structured data.

**GraphSAGE (Hamilton et al., 2017):**  A framework for inductive node embedding that uses node features to learn an embedding function that is able to generalize unseen nodes. By integrating node features into the learning algorithm, the model captures the topological structure of each node's neighborhood.

**APPNP (Gasteiger et al., 2022):**  This model uses GCN and PageRank to develop a framework called personalized propagation of neural predictions (APPNP), which utilizes a large adjustable neighborhood for classification and can be integrated with any neural network.

**SGC (Wu et al., 2019):** Motivated by the inherent complexity of GCNs, this model simplifies them by removing non-linearities between layers, reducing the function to a single linear transformation.

**FairGNN (Dai & Wang, 2021):** This model is designed to mitigate bias in GNNs while preserving high node classification accuracy by utilizing graph structures and minimal sensitive information.

**FairVGNN (Wang et al., 2022):** Motivated by observation of sensitive information leakage in GNNs, FairVGNN creates fair node features by automatically detecting and masking sensitive-correlated features, taking the variation of correlations after feature propagation into account.

**FairSIN (Yang et al., 2024):** Current methods address GNNs bias in predictions due to sensitive attributes by filtering out sensitive information. This has the potential of filtering out non-sensitive feature information, resulting in a trade-off between predictive performance and fairness. To overcome this, FairSIN incorporates Fairness-facilitating Features (F3) into node features or representations prior to message passing, with the goal of statistically neutralizing bias while retaining non-sensitive information.

### B.2 Baseline attacks

**TDGIA (Zou et al., 2021):** An injection attack based on topological vulnerability implements a defective edge strategy to identify the original nodes for connection with injected nodes. It then formulates a smooth feature optimization to generate features for the injected nodes.

**FA-GNN (Hussain et al., 2022):** This study highlighted the susceptibility of GNN models to adversarial fairness attacks by empirically identifying several edge injection strategies, which could compromise GNN fairness with only minimal impact on utility.

## C  Dataset specifics

Table 4: Statistics of the datasets. Adapted from Luo et al. (2024)

| Dataset | Pokec-z | Pokec-n | DBLP |
|---|---|---|---|
| # of nodes | 67,796 | 66,569 | 20,111 |
| # of edges | 617,958 | 517,047 | 57,508 |
| feature dim. | 276 | 265 | 2530 |
| # of labeled nodes | 10,262 | 8,797 | 3,196 |
| homophily rate | 0.95 | 0.96 | 0.79 |
| minority rate | 0.35 | 0.29 | 0.18 |

## D  Synthetic dataset configuration

### D.1  Graph generation

First, the number of nodes $N$ was set, and a proportion $f_m$ of these nodes was randomly assigned to the minority group, while the remainder formed the majority group. Each node received an activity score derived from a power-law distribution, parameterized separately for the majority ($\gamma_M$) and minority ($\gamma_m$) groups, reflecting their differential levels of connectivity potential.

Edges were created iteratively until the desired edge density $d$ was achieved. For each iteration, a source node $v_i$ was selected according to its activity level, and a target node $v_j$ was chosen with a probability proportional to the product of its in-degree $k_j^{\text{in}}$ and the homophily $h_{ij}$ between the source and target. The homophily values $h_{MM}$ and $h_{mm}$ governed the likelihood of intra-group connections for the majority and minority groups, respectively. The expected number of edges in the graph was scaled by the parameter $d$, ensuring that the total number of edges approximated $d \times N \times (N-1)$.

## D.2 Experimental settings

Multiple combinations of homophily rates and minority fractions were explored. Exploration ranges are based on the ranges used in the original paper (Espín-Noboa et al., 2022), as well as explorations by Ferrara et al. (2022).

The homophily values among majorities $h_{MM}$ and minorities $h_{mm}$ were sampled from the set $\{0.0, 0.1, 0.2, \ldots, 1.0\}$, while the fraction of minority nodes $f_m$ was set to $\{0.1, 0.3, 0.5\}$. For each combination of these parameters, networks were generated and analyzed under fixed conditions.

The following parameters were held constant across all experiments. The number of nodes was set to $N = 2000$, and the edge density $d$ was maintained at 0.0015, ensuring sparse graphs suitable for modeling real-world networks. The node activity followed a power-law out-degree distribution with parameters $\mathrm{plo}_M = \mathrm{plo}_m = 2.5$, reflecting heterogeneous connectivity patterns across groups. The number of feature dimensions needed for the proxy task was set to $n = 100$. Finally, all NIFA attacks were conducted using a perturbation rate of 1%, as per the original paper.

# E Results from the original paper

Table 5: Original authors' attack performance of NIFA on different victim GNN models. The results are reported in percentage (%). Adapted from Luo et al. (2024).

| | | Pokec-z | | | Pokec-n | | | DBLP | | |
| --- | --- | --- | --- | --- | --- | --- | --- | --- | --- | --- |
| | | Accuracy | $\Delta_{SP}$ | $\Delta_{EO}$ | Accuracy | $\Delta_{SP}$ | $\Delta_{EO}$ | Accuracy | $\Delta_{SP}$ | $\Delta_{EO}$ |
| GCN | Before | $71.22 \pm 0.28$ | $7.13 \pm 1.21$ | $5.10 \pm 1.28$ | $70.92 \pm 0.66$ | $0.88 \pm 0.62$ | $2.44 \pm 1.37$ | $95.88 \pm 1.61$ | $3.84 \pm 0.34$ | $1.91 \pm 0.75$ |
| | After | $70.50 \pm 0.30$ | $\mathbf{17.36 \pm 1.16}$ | $\mathbf{15.59 \pm 1.08}$ | $70.12 \pm 0.37$ | $\mathbf{10.10 \pm 2.10}$ | $\mathbf{9.85 \pm 1.97}$ | $93.37 \pm 1.48$ | $\mathbf{13.49 \pm 2.83}$ | $\mathbf{20.33 \pm 3.82}$ |
| GraphSAGE | Before | $70.79 \pm 0.62$ | $4.29 \pm 0.84$ | $3.46 \pm 1.12$ | $68.77 \pm 0.34$ | $1.65 \pm 1.31$ | $2.34 \pm 1.04$ | $96.58 \pm 0.29$ | $4.27 \pm 1.09$ | $2.78 \pm 0.91$ |
| | After | $70.05 \pm 1.25$ | $\mathbf{6.20 \pm 1.63}$ | $\mathbf{4.20 \pm 1.77}$ | $68.93 \pm 1.19$ | $\mathbf{3.32 \pm 1.88}$ | $\mathbf{3.56 \pm 1.91}$ | $93.92 \pm 0.74$ | $\mathbf{10.16 \pm 2.24}$ | $\mathbf{16.65 \pm 3.30}$ |
| APPNP | Before | $69.79 \pm 0.42$ | $6.83 \pm 1.25$ | $5.07 \pm 1.26$ | $68.73 \pm 0.64$ | $3.39 \pm 0.28$ | $3.71 \pm 0.28$ | $96.58 \pm 0.38$ | $3.98 \pm 1.18$ | $2.20 \pm 1.08$ |
| | After | $67.83 \pm 0.70$ | $\mathbf{18.44 \pm 1.41}$ | $\mathbf{16.85 \pm 1.50}$ | $67.90 \pm 0.76$ | $\mathbf{13.47 \pm 3.22}$ | $\mathbf{13.52 \pm 3.56}$ | $92.46 \pm 0.94$ | $\mathbf{13.88 \pm 3.20}$ | $\mathbf{20.20 \pm 4.25}$ |
| SGC | Before | $69.09 \pm 0.99$ | $7.28 \pm 1.50$ | $5.45 \pm 1.42$ | $66.95 \pm 1.69$ | $2.74 \pm 0.85$ | $3.21 \pm 0.78$ | $96.53 \pm 0.48$ | $4.70 \pm 1.26$ | $3.11 \pm 1.24$ |
| | After | $67.83 \pm 0.70$ | $\mathbf{17.65 \pm 1.01}$ | $\mathbf{16.09 \pm 1.06}$ | $66.72 \pm 1.21$ | $\mathbf{10.59 \pm 2.40}$ | $\mathbf{10.67 \pm 2.61}$ | $92.56 \pm 1.09$ | $\mathbf{13.88 \pm 3.37}$ | $\mathbf{20.25 \pm 4.44}$ |
| FairGNN | Before | $68.75 \pm 1.12$ | $1.89 \pm 0.63$ | $1.51 \pm 0.47$ | $69.41 \pm 0.66$ | $1.42 \pm 0.35$ | $2.32 \pm 0.57$ | $93.12 \pm 1.23$ | $1.95 \pm 0.99$ | $3.09 \pm 1.81$ |
| | After | $69.38 \pm 2.07$ | $\mathbf{5.71 \pm 2.52}$ | $\mathbf{4.22 \pm 1.89}$ | $69.97 \pm 0.42$ | $\mathbf{6.13 \pm 5.81}$ | $\mathbf{6.33 \pm 5.77}$ | $92.56 \pm 1.49$ | $\mathbf{5.89 \pm 2.52}$ | $\mathbf{10.48 \pm 3.82}$ |
| FairVGNN | Before | $68.57 \pm 0.45$ | $3.79 \pm 0.51$ | $2.59 \pm 0.59$ | $67.77 \pm 1.00$ | $1.90 \pm 1.23$ | $3.10 \pm 1.20$ | $95.18 \pm 0.54$ | $1.90 \pm 0.52$ | $2.91 \pm 1.05$ |
| | After | $67.65 \pm 0.38$ | $\mathbf{11.01 \pm 2.79}$ | $\mathbf{9.28 \pm 2.87}$ | $65.74 \pm 1.42$ | $\mathbf{3.51 \pm 1.51}$ | $\mathbf{3.65 \pm 1.56}$ | $91.56 \pm 1.13$ | $\mathbf{7.96 \pm 1.49}$ | $\mathbf{13.57 \pm 2.57}$ |
| FairSIN | Before | $67.33 \pm 0.22$ | $1.73 \pm 1.49$ | $2.61 \pm 1.44$ | $67.18 \pm 0.30$ | $0.39 \pm 0.89$ | $2.40 \pm 1.02$ | $94.72 \pm 0.62$ | $0.23 \pm 0.15$ | $0.45 \pm 0.16$ |
| | After | $66.55 \pm 0.44$ | $\mathbf{9.48 \pm 2.62}$ | $\mathbf{10.39 \pm 1.06}$ | $66.20 \pm 0.12$ | $\mathbf{11.82 \pm 0.75}$ | $\mathbf{14.58 \pm 0.22}$ | $92.46 \pm 0.32$ | $\mathbf{10.90 \pm 2.12}$ | $\mathbf{23.65 \pm 7.77}$ |

Table 6: Original authors' accuracy and fairness performance of attack launched by the different attackers. The results are reported in percentage (%). Adapted from Luo et al. (2024)

| | Pokec-z | | | Pokec-n | | | DBLP | | |
| --- | --- | --- | --- | --- | --- | --- | --- | --- | --- |
| | Accuracy | $\Delta_{SP}$ | $\Delta_{EO}$ | Accuracy | $\Delta_{SP}$ | $\Delta_{EO}$ | Accuracy | $\Delta_{SP}$ | $\Delta_{EO}$ |
| Clean | $71.22 \pm 0.28$ | $7.13 \pm 1.21$ | $5.10 \pm 1.28$ | $70.92 \pm 0.66$ | $0.88 \pm 0.62$ | $2.44 \pm 1.37$ | $95.88 \pm 1.61$ | $3.84 \pm 0.34$ | $1.91 \pm 0.75$ |
| AFGSM | $67.01 \pm 0.24$ | $3.07 \pm 1.67$ | $3.45 \pm 0.22$ | $68.21 \pm 0.23$ | $5.35 \pm 0.15$ | $5.68 \pm 0.14$ | $95.38 \pm 0.30$ | $5.44 \pm 3.48$ | $2.78 \pm 0.41$ |
| TGDIA | $62.20 \pm 0.04$ | $1.66 \pm 0.16$ | $0.77 \pm 0.10$ | $63.57 \pm 0.08$ | $7.28 \pm 0.35$ | $6.95 \pm 0.34$ | $93.42 \pm 0.29$ | $0.93 \pm 0.70$ | $1.82 \pm 0.87$ |
| G$^2$A2C | $39.41 \pm 0.94$ | $6.89 \pm 0.91$ | $6.11 \pm 0.48$ | $34.30 \pm 1.71$ | $2.23 \pm 1.40$ | $3.76 \pm 1.04$ | $86.28 \pm 0.25$ | $4.21 \pm 0.66$ | $3.80 \pm 0.42$ |
| FA-GNN | $69.80 \pm 0.48$ | $6.62 \pm 1.21$ | $8.67 \pm 1.28$ | $70.80 \pm 0.97$ | $2.64 \pm 0.76$ | $3.45 \pm 0.54$ | $95.48 \pm 0.48$ | $3.32 \pm 1.65$ | $8.74 \pm 1.23$ |
| FATE | - | - | - | - | - | - | $94.87 \pm 0.41$ | $3.62 \pm 1.49$ | $2.12 \pm 1.01$ |
| G-FairAttack | - | - | - | - | - | - | $95.12 \pm 0.38$ | $6.80 \pm 0.59$ | $2.94 \pm 1.10$ |
| NIFA | $70.50 \pm 0.30$ | $\mathbf{17.36 \pm 1.16}$ | $\mathbf{15.59 \pm 1.08}$ | $70.12 \pm 0.37$ | $\mathbf{10.10 \pm 2.10}$ | $\mathbf{9.85 \pm 1.97}$ | $93.37 \pm 1.48$ | $\mathbf{13.49 \pm 2.83}$ | $\mathbf{20.33 \pm 3.82}$ |

# F Exploratory analysis

This section presents an exploratory analysis investigating how varying levels of homophily and minority group fractions influence fairness outcomes under the NIFA attack. As graph homophily governs the tendency of nodes to connect with others sharing the same attributes, changes in homophily may systematically affect

how injected nodes propagate bias. We assess whether fairness metrics—specifically $\Delta_{SP}$ and $\Delta_{EO}$—respond predictably to these structural changes and whether this response interacts with the proportion of minority group members.

The analysis combines homophily settings (ranging from highly heterophilic to highly homophilic) with varying minority group fractions ($f_m \in 0.1, 0.3, 0.5$). This design allows us to evaluate not only the independent effects of each variable but also potential interaction effects between minority size and structural cohesion.

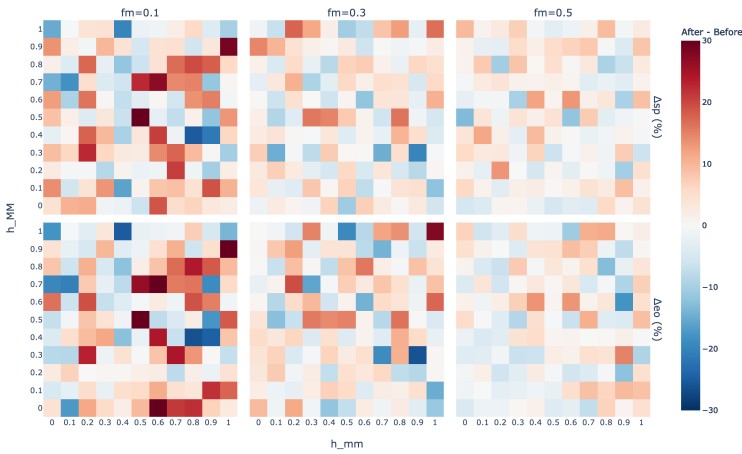

Figure 4: Impact of homophily and minority fraction on fairness metrics under NIFA.

As shown in Figure 4, no clear monotonic trend was observed in $\Delta_{SP}$ or $\Delta_{EO}$ as a function of homophily level alone. In particular, fairness degradation did not increase or decrease consistently across the homophily spectrum. This suggests that while homophily affects fairness in some configurations (e.g., mixed or extreme values), its relationship with fairness degradation is not linear or straightforward.

In contrast, the minority fraction demonstrated a more consistent effect. Graphs with lower minority representation ($f_m = 0.1$) exhibited significantly greater shifts in fairness metrics after the NIFA attack, particularly in settings with moderate to high homophily. This aligns with the intuition that when minority nodes are sparsely represented, targeted node injections can more easily skew group-level statistics by amplifying the representation of the majority group. This suggests that group imbalance exacerbates vulnerability to fairness attacks, even when homophily levels are held constant.

Although preliminary, these findings highlight the need for deeper investigation into how structural factors (e.g., community structure, degree distribution) and demographic variables (e.g., group size imbalance) jointly affect fairness robustness. Future work could explore more diverse synthetic graph generators, introduce continuous variations in homophily and group sizes, and test interactions with different fairness-aware training objectives.

In summary, this analysis provides initial evidence that minority representation plays a larger and more systematic role than homophily in determining fairness vulnerability under node injection attacks. However, more granular and statistically powered studies are required to draw firm conclusions about these interactions.

