# OpenReview forum: "Reassessing Fairness: A Reproducibility Study of NIFA’s Impact on GNN Models"
_TMLR — Accepted by TMLR_

### Review · Reviewer_JxUV · 2025-03-27

**Summary Of Contributions:**

The study aims to reproduce the fairness degradation effects of the Node Injection-based Fairness Attack (NIFA) on Graph Neural Networks (GNNs), and to extend its evaluation to multi-class sensitive attributes and varying graph homophily levels.
The study evaluates three specific claims of NIFA, a gray-box fairness attack on GNNs, by replicating the original experiments using three real-world datasets (Pokec-z, Pokec-n, DBLP) and four classical GNN models (GCN, GraphSAGE, APPNP, and SGC), as well as fairness-aware models (FairGNN, FairVGNN, and FairSIN). The three claims evaluated are: NIFA's fairness degradation in GNNs while maintaining negligible utility loss, NIFA state-of-the-art performance with scalability and minimal detectability, and NIFA's outperformance over existing fairness and utility attack baselines across multiple datasets. NIFA degrades fairness by injecting synthetic nodes during training, leveraging two key principles: uncertainty maximization using Bayesian GNNs, and increasing homophily by linking injected nodes within the same sensitive group. Features of these nodes are optimized via a multi-objective loss function balancing classification accuracy and fairness degradation.

This paper also extends the original scope by adapting NIFA to multi-class sensitive attributes using One-vs-All and One-vs-One generalizations of fairness metrics, and by evaluating its effectiveness across synthetic graphs with different homophily levels. The experiments also include comparisons with fairness (FA-GNN) and utility-based (TDGIA) attack baselines. Results are averaged over five runs using a consistent hardware setup, with minor code adjustments due to dependency mismatches. The study assesses both fairness (∆SP and ∆EO) and utility (accuracy), analyzing pre- and post-attack differences to determine NIFA’s robustness and generalizability across configurations.

The paper confirms that NIFA significantly degrades fairness in classical GNNs with minimal utility loss, though its superiority over baseline attacks is inconclusive. The attack remains effective with multi-class sensitive attributes and under varying homophily, though highly homophilic graphs show increased metric variability, suggesting structural sensitivity.

**Audience:**

Yes

**Broader Impact Concerns:**

Since this is a reproducibility study, as I indicated above, the main concern is that given the limited set of attacks that are evaluated, the paper can be mistaken as a full validation of NIFA. The details of the atacks studied should be clarified further in the abstract and introduction. This could limit the quality of this paper's broader impact.

**Claims And Evidence:**

No

**Requested Changes:**

I would suggest to clarify some details of the study. For instance, indicating the specific obstacles in reproducing the remaining four baseline attacks (e.g., AFGSM, G2A2C, FATE, G-FairAttack) or completing these analysis will provide a more thorough analysis. A time constraint does not sound like a valid scientific justification. Also, it could be helpful to clarify whether pre-trained models or author-provided outputs could be used as a proxy to compare NIFA’s performance.

Other information that could be useful for this analysis are, first, the implementation details or configurations that differed between the original and reproduced fairness-aware models and, second, how sensitive are the fairness metrics (∆SP and ∆EO) to hyperparameters or random seed initialization in these models.

**Strengths And Weaknesses:**

The most notable strength of this reproducibility study is that it goes beyond simple reproduction by extending the evaluation to multi-class sensitive attributes and different graph homophily levels, offering deeper insights into NIFA’s generalizability and robustness in more realistic and diverse scenarios. Another strength of this paper is that it provides a well-documented methodology, including detailed descriptions of dataset handling, model adjustments, evaluation metrics, and attack configurations, which enhances the reproducibility and credibility of their findings. Thus, the paper carefully differentiates between partially and fully reproduced claims, discussing potential reasons for discrepancies, such as differences in code availability or experimental environments.

The paper also has limitations. Most notably, only two of the six baseline attacks from the original study were reproduced (TDGIA and FA-GNN) due to time constraints. This limits the strength of the comparative evaluation and weakens the assessment of NIFA’s relative performance. An additional observation is that, since some discrepancies for the fairness-aware models and the  running fairness-aware models may be due to differences in code implementation. Thus, as currently structure, the paper does not fully resolve the divergences due to methodological consideration.

---

> ### Author Response · Authors · 2025-05-08
>
> Thank you for your detailed and thoughtful feedback. While we were able to address some of your recommendations, some could not be fully implemented due to the practical limitations described earlier.
>
> Limited baseline reproduction: We acknowledge this limitation. While time and technical constraints played a role, the main challenge was the lack of accessible or compatible implementations for the remaining baselines. To avoid placing undue criticism on the original authors, we chose not to dwell on these difficulties in detail. Instead, given the scope of our project and the constraints we faced, we decided to focus our efforts on extending the original work. We’ve slightly reframed our explanation in Sections 2 and 5.1 to reflect this more clearly.
>
> Section 6.1 now explains that only classical GNNs were directly supported in the original repository. Fairness-aware models were sourced independently, and we note that these implementation differences likely contributed to the observed metric discrepancies.
>
> We have clarified key aspects of the study based on your feedback. Specifically, we now discuss the sensitivity of fairness metrics to random initialization (Sections 5.2 and 6.1), clearly position the paper as a partial reproduction and extension (Abstract, Introduction, and Scope), and added a Broader Impact paragraph (end of Section 6.1) to highlight the ethical significance of studying fairness vulnerabilities.

---

### Review · Reviewer_h4eE · 2025-03-28

**Summary Of Contributions:**

This paper aims to check the reproducibility of the NiFA algorithm (attack on the fairness of GNN). The studies address the NIFA paper's claims thoroughly and add some experiments concerning the influence of homophily on graphs. This paper validates the NIFA's claims, except for some setups, such as fairness-aware models. Multi-class and the influence of homophily are examined carefully. It implies the susperiority of NiFA compared to others, and the influence of homophily cannot be little. Furthermore, the paper discusses the reproducibility issues of NIFA in detail.

**Audience:**

Yes

**Broader Impact Concerns:**

There is no broad impact statement. However, the fairness algorithm is related to ethical issues such as fairness in decision-making. Also, the topic of NiFA can be related to the security issue.

**Claims And Evidence:**

Yes

**Requested Changes:**

I suggest the following issues.

1. The analysis of the influence of homophily is too short. More insightful discussion or analysis can strengthen the paper. What's the reason the case of Hete /Homo(0.36) shows the high value in Fig3.
2. More datasets are required to generalize the NIFA's performance. Therefore, many datasets should be examined if possible.
3. The scope of the task should be clarified kindly. Is the task a node classification? Is the edge classification valid in this paper?

**Strengths And Weaknesses:**

Pros: This paper is well-written and clear. The message is obvious, and the discussion looks sound.
Cons: This paper provides no new algorithm and only deals with reproducibility issues. I'm afraid this paper's message is not big on ML society. It can be a good paper if the reproducibility issue is considered an important fold.

---

> ### Author Response · Authors · 2025-05-08
>
> Thank you for your detailed and constructive feedback. We took your comments to heart and did the following:
>
> 1. We expanded Section 5.2 with a hypothesis explaining the spike in the Hete/Homo configuration and extended our analysis in the Appendix with a heatmap exploring interactions between homophily and minority fraction.
>
> 2. While we did not add new real-world datasets, we addressed generalizability by systematically varying homophily and group representation in synthetic graphs. This is now discussed more explicitly in the results and discussion.
>
> 3. We clarified in Section 3.3 that our study focuses solely on node classification, in line with the original NIFA framework.

---

### Review · Reviewer_nuQd · 2025-04-08

**Summary Of Contributions:**

The paper examines the fairness properties of graph neural networks (GNNs), in particular, the node injection-based fairness attack (NIFA) recently proposed by Luo et al. (2024). The paper aims to reproduce the main claims made by Luo et al., most of which are corroborated by this paper. They also extend the work of Luo et al. to account for multi-class sensitive attributes while also investigating the role of homophily in NIFA. They find that NIFA is effective even in the presence of multi-class sensitive attributes. Moreover, they find that homophily correlates with how effective NIFA is.

**Audience:**

Yes

**Broader Impact Concerns:**

No broader impact concerns.

**Claims And Evidence:**

Yes

**Requested Changes:**

Two minor points:

- Typo on page 1: "ampl amplefies"
- Eqs. (3) and (4) should have absolute values, right?

**Strengths And Weaknesses:**

The main contribution of the paper lies in reproducing and extending the recent paper by Luo et al. Fairness in the context of GNNs is an important research topic, and reproducibility studies are certainly valuable in guiding future research. The paper finds that the key claims made by Luo et al. are to a large extent accurate, although it also reports certain inconsistencies. Those findings are certainly relevant for the community going forward, and deserve to be highlighted. Furthermore, the paper conducts some further experiments concerning natural extensions to the paper of Luo et al. Understanding how effective NIFA is in the presence of multi-class sensitive attributes is a natural next step. I also found the study relating the effectiveness of NIFA to the homophily level interesting. Taken as a whole, the paper contains some interesting results that deserve to be highlighted. The paper is well-written, providing adequate background and accurately placing its contributions in the context of the existing literature.

On the negative side, it is unclear whether the results are significant enough for this venue. There are very few challenges in reproducing the results of Luo et al. Reproducibility studies are more impactful when examining well-established and influential papers; the result of Luo et al. is very recent and not well-cited, so it's unclear whether this paper will have an impact going forward.

---

> ### Author Response · Authors · 2025-05-08
>
> Thank you for your thoughtful and constructive feedback. We apologize for the delayed response and greatly appreciate the time you took to review our work.
>
> Typo (“ampl amplefies”): Corrected in the abstract.
>
> Equations (3) and (4): Updated to include absolute values for consistency with standard fairness definitions.
>
> Multi-class and homophily extensions: We’ve emphasized these contributions more clearly throughout the abstract, introduction, and results sections. We also expanded the homophily analysis slightly in Section 5.2 and the Appendix.
>
> Limited impact due to NIFA’s recency: We agree and now clearly position the paper as a scoped reproducibility and extension study, rather than a comprehensive validation.

---

### Author Response · Authors · 2025-04-25
**Follow-Up on TMLR Submission**

Dear Sheng Li and Reviewers,

First of all, we would like to sincerely thank you for the time and thoughtful feedback you have provided on our submission. We truly appreciate the effort you have taken to read our work and share detailed and constructive suggestions for improvement.

We must also express our regret for not following up within the expected timeframe. After submitting to TMLR as part of our university's programme—with the primary aim of participating in the ML Reproducibility Challenge (MLRC) 2025—we unfortunately underestimated the time required to adequately revise your comments. As we returned to our ongoing studies and other academic commitments, we were not able to dedicate the necessary attention to respond promptly.

That said, we have taken your comments seriously and plan to revise the paper to incorporate your suggestions, particularly the minor changes that are well within reach. While we are not in a position to conduct more extensive additional studies or benchmarking at this time, we hope to update the paper to better reflect the clarity and quality standards you’ve encouraged through your feedback.

We understand that the scope and impact of the work may not fully align with TMLR’s expectations, as one reviewer rightly noted. Nonetheless, we are grateful for the opportunity to have submitted and received such valuable feedback.

Thank you again for your time and understanding.

---

> ### Comment · Action_Editor_5RXv · 2025-04-25
>
> Dear Authors,
>
> Thanks for letting us know your plan. We are glad that you found the comments are constructive and helpful.
>
> Best,
>
> AE

---

### Author Response · Authors · 2025-05-08
**Resubmission**

Thank you again for your thoughtful and constructive feedback on our submission. We sincerely appreciate the time and care you dedicated to reviewing our work. We regret not responding within the expected timeframe.

We have now carefully addressed your comments and revised the paper accordingly. While we were not in a position to conduct more extensive benchmarking, we have clarified our scope, improved the discussion of limitations, and incorporated your suggestions to the extent possible. We’ve also emphasized that this is a partial reproduction and extension study, rather than a full validation of NIFA.

We have resubmitted the updated version and hope it better reflects the clarity and standards you encouraged through your feedback.

---

### Author Response · Authors · 2025-06-13
**Decision on Submission**

Dear action editor,

Thank you again for your thoughtful and constructive feedback. Although our response to the initial round of reviews was somewhat delayed, we took your comments seriously and have carefully revised the paper to reflect your suggestions.

While we were not in a position to conduct more extensive benchmarking, we clarified the scope of our contribution, improved the discussion of limitations, and emphasized that this is a partial reproduction and extension of NIFA rather than a full validation. We resubmitted the revised paper and hope the changes align with the clarity and standards you encouraged.

Given the upcoming MLRC decision deadline, we would be grateful if you could consider rendering a decision in time for the June 20th cutoff. We remain genuinely enthusiastic about participating in the MLRC challenge and appreciate the opportunity to contribute to this forum.

Kind regards,

The authors

---

### Decision · Action_Editor_5RXv · 2025-06-06

**Recommendation:** Accept as is

**Audience:**

Yes

**Audience Explanation:**

Safety of graph neural networks is an emerging research topic. Some researchers in TMLR's audience would be interested in learning the reproducibility study of this work.

**Claims And Evidence:**

Yes

**Claims Explanation:**

This paper presents a reproducibility study of the node injection-based fairness attack (NIFA) method proposed by Luo et al. in 2024. The authors reproduced and evaluated NIFA across multiple datasets and GNN architectures, which confirmed that NIFA consistently degrades fairness.